# Alectinib in Early-Stage Anaplastic Lymphoma Kinase-Positive Non-Small Cell Lung Cancer: Current Evidence and Future Challenges

**DOI:** 10.3390/cancers16142610

**Published:** 2024-07-22

**Authors:** Diego Luigi Cortinovis, Alessandro Leonetti, Alessandro Morabito, Luca Sala, Marcello Tiseo

**Affiliations:** 1Division of Medical Oncology, Fondazione IRCCS San Gerardo dei Tintori, 20900 Monza, Italy; diegoluigi.cortinovis@irccs-sangerardo.it; 2School of Medicine and Surgery, University of Milano Bicocca, 20126 Milan, Italy; luca.sala@irccs-sangerardo.it; 3Medical Oncology Unit, University Hospital of Parma, 43126 Parma, Italy; mtiseo@ao.pr.it; 4Thoracic Medical Oncology, Istituto Nazionale Tumori, IRCCS “Fondazione G. Pascale”, 80131 Naples, Italy; a.morabito@istitutotumori.na.it; 5Department of Medicine and Surgery, University of Parma, 43126 Parma, Italy

**Keywords:** non-small cell lung cancer, anaplastic lymphoma kinase, disease-free survival, targeted therapies, adjuvant therapy, alectinib

## Abstract

**Simple Summary:**

The tyrosine kinase inhibitor alectinib is currently the first-line treatment for advanced *ALK*-positive non-small cell lung cancer. The aim of this commentary is to highlight data from clinical trials, case reports, and case series evaluating alectinib in patients with early-stage *ALK*-positive non-small cell lung cancer in the adjuvant and perioperative setting. The results of the commentary suggest that alectinib could be the new adjuvant option for stage IB-IIIA *ALK*-positive NSCLC based on the phase III ALINA trial and, in the near future, the results of the phase II ALNEO and NAUTIKA1 trials, evaluating alectinib in the neoadjuvant/perioperative setting, which could further modify the current therapeutic strategy.

**Abstract:**

Background: Targeted therapies changed the treatment of advanced oncogene-addicted non-small cell lung cancer and could also improve outcomes in resectable disease. Results: The ALINA trial evaluated the clinical benefit of adjuvant alectinib compared with standard chemotherapy and met the primary endpoint with a significant increase in disease-free survival at 2 years among anaplastic lymphoma kinase positive patients with stage IB-IIIA disease; two phase II trials (ALNEO and NAUTIKA1) are currently evaluating perioperative treatment with alectinib, and the results of the case reports published to date are encouraging. Conclusion: In resectable anaplastic lymphoma kinase-positive lung cancer, adjuvant alectinib represents the new standard of care and could soon be used in perioperative treatment.

## 1. Introduction

Lung cancer is one of the most common cancers worldwide, 80–85% of which is non-small cell lung cancer (NSCLC) [1]. Alterations in the anaplastic lymphoma kinase (*ALK*) gene are one of the most common driver mutations detected among Western and Asian patients with lung adenocarcinoma in 4% of cases [2]. At the time of the development of first-generation tyrosine kinase inhibitors (TKIs), the 4-year overall survival (OS) for advanced *ALK*-rearranged NSCLC was 56.6% [3]. Alectinib, brigatinib, and lorlatinib (second and third-generation *ALK* inhibitors) showed superior progression-free survival (PFS) and safety compared with crizotinib in the primary treatment of *ALK*-positive NSCLC in phase III trials [4,5,6]. Following the advent of novel *ALK*-TKIs, the median PFS exceeded 30 months, and the 5-year OS rate reached 60% in this population [7]. Alectinib, brigatinib, and lorlatinib are the current standard first-line treatment of *ALK*-rearranged NSCLC.

Targeted therapies may also improve outcomes in patients with resectable oncogene-addicted disease. In this context, the phase III ADAURA trial demonstrated the significant benefit of adjuvant osimertinib versus placebo in terms of disease-free survival (DFS) in patients with stage IB-IIIA epidermal growth factor receptor (EGFR) mutation-positive NSCLC, regardless of the use of adjuvant chemotherapy. These results translated into a significant improvement in OS, shifting the treatment paradigm of radically resected EGFR-mutated NSCLC [8,9]. In resectable *ALK*-rearranged NSCLC, ongoing an ALCHEMIST trial (NCT02194738) is evaluating adjuvant crizotinib at 250 mg twice daily versus observations in stage IB-IIIA NSCLC patients after standard care chemotherapy and/or radiotherapy.

## 2. Alectinib in Early-Stage NSCLC: ALINA Trial

ALINA (NCT03456076) was a global, randomized, open-label, phase III trial evaluating the clinical benefit of adjuvant alectinib compared to standard chemotherapy in patients with *ALK*-positive NSCLC in stage IB (tumors ≥ 4 cm), II, or IIIA (according to the 7th edition of the Cancer Staging Manual of the American Joint Committee on Cancer and the Union for International Cancer Control). In total, 257 patients were included in the study. Patients were randomized in a 1:1 ratio to receive oral alectinib at 600 mg twice daily (until disease recurrence or for up to 24 months) or standard platinum-based chemotherapy. The formal crossover design was not incorporated into the study, and subsequent treatments were left to the discretion of the investigators. The primary endpoint was DFS in stage II or IIIA disease; then, in stage IB-IIIA disease, the intention-to-treat (ITT) population was tested hierarchically. Secondary endpoints were central nervous system (CNS)-DFS, OS, and safety.

The demographic and clinical characteristics of patients at baseline were generally well-balanced between the two study groups and reflective of the available demographic data on patients with *ALK*-positive NSCLC. The only characteristics unbalanced in favor of the alectinib group were female sex (57.7% versus 46.5%) and those who never smoked (64.6% versus 55.1%). After a median follow-up of approximately 28 months, the study met the primary endpoint. The 2-year DFS rate in stage II-IIIA disease was 93.8% in the alectinib group versus only 63.0% in the control group (the hazard ratio [HR] for disease recurrence or death was 0.24, with a 95% confidence interval [CI] 0.13–0.45, *p* < 0.0001) and 93.6% versus only 63.7%, respectively, in the ITT population (HR 0.24, 95% CI 0.13–0.43, *p* < 0.0001). The DFS benefit of adjuvant alectinib was generally consistent across all subgroups, including stage IB, with an HR of 0.21. Of note, adjuvant alectinib significantly improved CNS DFS, with an HR of 0.22 (95% CI 0.08–0.58) (Table 1).

A total of 15 patients (11.5%) in the alectinib group had disease recurrence compared to 49 (38.6%) in the chemotherapy arm. The most common site of recurrence was the brain, reported in four patients in the alectinib group and 14 in the chemotherapy group. Data on OS are not yet available.

No unexpected safety findings were observed. In total, 98.4% of patients in the alectinib group and 93.3% of chemotherapy group patients reported at least one adverse event. The most reported adverse events were increased creatine kinase levels (43.0%) and constipation (42.2%) for the alectinib group compared to nausea (72.5%) and decreased appetite (29.2%) for the chemotherapy group. Treatment-related grade 3 or 4 adverse events occurred in 18.0% and 27.5% of patients, respectively. Considering serious adverse events, the incidence of pneumonitis induced by adjuvant alectinib therapy was 2.3%, similar to 2% for first-line treatment. Due to adverse events, in the alectinib group, 33 (25.8%) patients had a dose reduction, and 7 (5.5%) patients had a dose interruption [10].

## 3. Future Perspectives

Considering the remarkable efficacy of *ALK*-TKIs in terms of their objective response rate (ORR) compared with chemotherapy in the metastatic setting, neoadjuvant or perioperative target therapy may be a promising strategy for *ALK*-positive patients [11,12]. In 2018, Zhang et al. described a case series of eleven pN2 *ALK*-positive NSCLC patients treated with neoadjuvant crizotinib. After neoadjuvant treatment, 10 patients had a partial response, and 1 patient had stable disease. Ten patients received an R0 resection, and 20% of these achieved a pathologic complete response (pCR) [13]. The phase II ARM trial (NCT03088930), evaluating the activity of 6-week oral neoadjuvant crizotinib in resectable stage IA-IIIA *ALK*+ NSCLC patients, closed due to difficulty with accrual. The phase II SAKULA trial, evaluating the activity of 12-week oral neoadjuvant ceritinib at 750 mg daily in resectable stage II-III *ALK*+ NSCLC patients, only enrolled seven patients all with stage IIIA disease; ORR was 100%, six patients underwent surgical resection and pCR was 28% [14]. To date, the regimens and timing of neoadjuvant/perioperative treatment with alectinib have not been well studied, and no study results have been published to date. The available literature is limited to a few case reports/case series with neoadjuvant crizotinib, ceritinib, and alectinib. Reports of neoadjuvant or perioperative treatment with alectinib achieved interesting results in terms of the pathologic response (Table 2) [15,16,17,18,19,20,21,22,23,24].

Two phase II trials are currently evaluating perioperative alectinib in *ALK*-positive NSCLC (Figure 1).

In potentially resectable stage III *ALK*-positive NSCLC patients (any T with N2, T4N0-1), the ongoing ALNEO, an Italian multicenter, open-label, single-arm, phase II trial, is evaluating 8-week neoadjuvant alectinib at 600 mg twice daily. After definitive surgery, patients receive adjuvant alectinib for 24 months. The primary endpoint is major pathologic response (MPR); secondary endpoints are pCR, ORR, event-free survival (EFS), DFS, OS, and adverse events [17]. The other ongoing study is NAUTIKA1, an umbrella trial investigating perioperative-targeted therapies in biomarker-selected patients, including alectinib in patients with resectable stage II-III *ALK*-positive NSCLC (according to the 8th edition of the Cancer Staging Manual of the American Joint Committee on Cancer and the Union for International Cancer Control). A preliminary analysis assessed data on the first five patients who underwent neoadjuvant alectinib followed by surgery. They all completed at least 8 weeks of neoadjuvant treatment, underwent surgery during the protocol-defined window, and had a complete resection with no delays or major complications [25].

## 4. Discussion

ALINA achieved the primary endpoint of DFS with alectinib versus standard chemotherapy in resected *ALK*-positive NSCLC. The primary endpoint was also achieved in the intention-to-treat population that included stage IB, a subgroup that could benefit from adjuvant alectinib when also considering that *ALK* rearrangement in stage I NSCLC is significantly associated with poor prognosis compared to *ALK*-negative patients [26,27]. A limitation of this study was the inclusion of stage IB patients with tumors larger than 4 cm without taking into account other risk factors for disease recurrence. Currently, spread through air spaces (STAS) in stage I NSCLC is a risk factor significantly associated with poor prognosis for both OS and DFS, regardless of tumor size, and STAS is frequently observed in *ALK*-positive NSCLC [28,29]. Visceral pleural invasion (VPI) is also a significant negative prognostic factor in stage IB NSCLC [30]. It is appropriate to speculate that patients with an *ALK*-positive stage IB NSCLC tumor smaller than 4 cm and concomitant risk factors such as STAS and VPI could also benefit from adjuvant treatment with alectinib. Although patient characteristics were generally well balanced between the two study groups, the alectinib group had a higher proportion of female patients and those who never smoked, which may have influenced the outcome by increasing the benefit attributed to alectinib over chemotherapy. DFS is a well-established efficacy endpoint in studies of adjuvant therapy in resectable NSCLC [31,32]. Recent data from adjuvant osimertinib in EGFR-positive resected NSCLC indicate the possibility that a broad DFS benefit with a highly effective TKI will result in an OS benefit in the adjuvant setting. *ALK*-positive NSCLC patients are at high risk of intracranial disease, which is associated with poor prognosis. Adjuvant alectinib may prevent or delay recurrence in the CNS based on ALINA trial results, and these findings are consistent with the intracranial efficacy of alectinib in advanced NSCLC [33].

The results of the ALINA study raise questions about the use of targeted agents as adjuvant therapy. First, the ALINA trial does not address the potential of adding chemotherapy to alectinib, an approach that may allow therapy intensification in selected groups of patients. This is in spite of the fact that ADAURA showed a DFS benefit favoring osimertinib over the placebo with/without adjuvant chemotherapy, regardless of disease stage [34].

Second, the most effective timing and duration of adjuvant-targeted therapy in resectable NSCLC have not yet been established. The TARGET trial will assess the efficacy and safety of 5 years of adjuvant osimertinib in patients with fully resected, EGFR-positive stage II-IIIB NSCLC [35]. Tailoring the duration of adjuvant treatment, with or without chemotherapy, according to the risk of disease recurrence may require further investigation. In the future, minimal residual disease (MRD) detected with circulating tumor DNA (ctDNA) in the perioperative setting may have both prognostic and predictive values for treatment choice [36,37]. Preliminary data in NSCLC support the use of ctDNA profiling as a method of measuring the risk of progression postoperatively and during treatment. In advanced NSCLC, ctDNA clearance during treatments has been associated with ORR, PFS, and OS. If validated, ctDNA could become a new surrogate endpoint of interest in adjuvant treatment studies to detect early disease recurrence when assessed longitudinally [38,39].

Third, an important practical consequence of ALINA and ADAURA is that molecular profiling using next-generation sequencing is useful for the therapeutic strategy already at diagnosis for potentially resectable tumors that require additional adjuvant or neoadjuvant systemic therapy. The significant benefit of adjuvant alectinib has made the early identification of tumors with ALK fusions even more important to avoid neoadjuvant chemo-immunotherapy treatments. The extension of NGS to all resectable NSCLC patients could broaden the application of targeted therapies to all actionable genetic alterations. As an example, in patients with stage IB-IIIA *RET* fusion-positive NSCLC previously treated with standard therapy, the ongoing phase III LIBRETTO-432 study is evaluating adjuvant selpercatinib versus a placebo [40].

Fourth, ALINA data are immature to define the subsequent treatment strategy in cases of disease recurrence. In the experimental arm, only 15 patients experienced disease recurrence, and subsequent treatments were determined by the investigator; subsequent treatments included *ALK*-TKI for 7 patients (alectinib or brigatinib), chemotherapy for 6 patients, immunotherapy for one patient and other anti-cancer therapy for one patient. The therapeutic strategy could theoretically be differentiated according to the time of recurrence, and a biopsy of the recurrence site would be mandatory to define the mechanism of resistance and the most appropriate subsequent treatment. If relapse occurs during 2 years of adjuvant therapy, it may be secondary to the development of mechanisms of resistance to alectinib. Lorlatinib has the most relevant data on advanced diseases with on-target resistance and is the best option in this case. In the case of off-target resistance, the choice should fall either to dual chemotherapy with platinum or pemetrexed chemotherapy or to enrollment into a clinical trial [41]. If relapse occurs after the end of adjuvant therapy, a rechallenge with alectinib should also be considered because the relapse could be due to the reactivation of drug-tolerant persister cells [42].

Lastly, the first case of neoadjuvant treatment with lorlatinib was recently published. A patient diagnosed with unresectable stage IIIA (cT1bN2M0) *ALK*-positive NSCLC underwent surgery to achieve a radical resection after neoadjuvant therapy with 3 months of lorlatinib and a pCR, which was confirmed by pathological analysis [43]. This result, in agreement with the most recent data from the CROWN study [44], also raises the question about the use of *ALK*’s latest generation inhibitors in the adjuvant and perioperative setting.

## 5. Conclusions

The ALINA trial was the first study to demonstrate a significant improvement in DFS with adjuvant alectinib in resected *ALK*-positive NSCLC. Based on the results of the ALINA study, alectinib has been shown to be an effective and tolerable adjuvant option for *ALK*-positive NSCLC, which may also replace chemotherapy and FDA-approved alectinib as a form of adjuvant treatment for *ALK*-positive non-small cell lung cancer on 18 April 2024, and EMA on 25 April 2024. Prospective studies need to be conducted to identify the most appropriate treatment duration of adjuvant-targeted therapies and the value of adding adjuvant chemotherapy in resectable NSCLC, which may be different depending on factors such as the molecular profile of the tumors or the presence of minimal residual disease. In the near future, the results of the ALNEO and NAUTIKA1 trials may further explore the role of neoadjuvant/perioperative alectinib in the treatment paradigm of *ALK*-positive NSCLC patients.

## Figures and Tables

**Figure 1 cancers-16-02610-f001:**
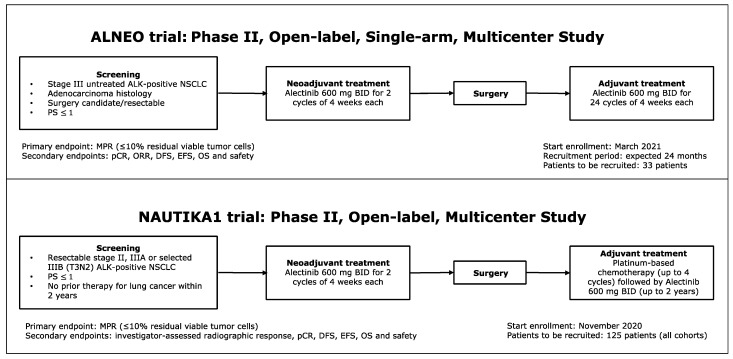
Schema for ALNEO trial and NAUTIKA1 trial. Abbreviations: *ALK* = anaplastic lymphoma kinase; BID = twice a day; DFS = disease-free survival; EFS = event-free survival; MPR = major pathologic response; NSCLC = non–small-cell lung cancer; ORR = objective response rate; OS = overall survival; pCR = pathological complete response; and PS = performance status.

**Table 1 cancers-16-02610-t001:** The 2-year disease-free survival rate among patients with stage II or IIIA disease, those in the intention-to-treat population, those with stage IB disease, and the 2-year central nervous system disease-free survival rate in the intention-to-treat population. Abbreviations: CI = confidence interval; CNS = central nervous system; and DFS = disease-free survival.

	Alectinib	Chemotherapy	Hazard Ratio	95% CI	*p* Value
2-year DFS rate stage II-IIIA	93.8%	63.0%	0.24	0.13–0.45	<0.001
2-year DFS rate stage IB-IIIA	93.6%	63.7%	0.24	0.13–0.43	<0.001
2-year DFS rate stage IB	92.3%	71.6%	0.21	0.02–1.84	
2-year CNS DFS rate stage IB-IIIA	98.4%	85.8%	0.22	0.08–0.58	

**Table 2 cancers-16-02610-t002:** Reported cases receiving neoadjuvant alectinib therapy. Abbreviations: BID = twice a day; CR = complete response; MPR = major pathological response; NA = not available; pCR = pathological complete response; PORT = postoperative radiotherapy; and PR = partial response.

Case	Author	Age (Year)	Gender	Stage	Neoadjuvant Treatment	Radiologic Response	Patologic Response	Adjuvant Treatment
1	Zhang et al. [15]	46	Male	cIIIB	Alectinib 600 mg BID for 8 weeks	PR	Non-MPR	NA
2	Yue et al. [16]	51	Male	cIIIA	Alectinib 600 mg BID for 6 weeks	PR	Non-MPR	PORT, Alectinib
3	Leonetti et al. [17]	62	Male	cIIIA	Alectinib 600 mg BID for 8 weeks	PR	MPR	Alectinib 600 mg BID for 24 months
4	Gu et al. [18]	67	Male	cIIIB	Alectinib 150 mg BID for 12 weeks	CR	MPR	NA
5	Hu et al. [18]	58	Female	cIIIA	Alectinib 600 mg BID for 8 weeks	PR	pCR	Alectinib
6	Sentana-Lledo et al. [20]	61	NA	cIIIA	Alectinib 600 mg BID for 6 weeks	PR	pCR	Alectinib 600 mg BID for 24 months
7	Sentana-Lledo et al. [20]	65	NA	cIIIA	Alectinib 600 > 450 mg BID for 12 weeks	PR	MPR	NA
8	Shi et al. [21]	51	Male	cIIB	Alectinib 600 mg BID for 30 weeks	PR	pCR	Alectinib
9	Shi et al. [21]	48	Male	cIIB	Alectinib 600 mg BID for 32 weeks	PR	pCR	Alectinib 600 mg BID for 24 months
10	Wang et al. [22]	41	Male	cIIIB	Alectinib 600 mg BID for 14 months	PR	pCR	Alectinib
11	Wang et al. [23]	64	Male	cIIIB	Alectinib 600 mg BID for 44 days	PR	pCR	NA
12	Wang et al. [24]	52	Female	cIIIB	Alectinib 600 mg BID for 10 weeks	PR	pCR	Alectinib

## Data Availability

No new data were created or analyzed in this study. Data sharing is not applicable to this article.

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
