# Peer review of "Alectinib in Early-Stage Anaplastic Lymphoma Kinase-Positive Non-Small Cell Lung Cancer: Current Evidence and Future Challenges"

_cancers, 2024, doi:10.3390/cancers16142610_

Round 1

Reviewer 1 Report

Comments and Suggestions for Authors

This manuscript addresses potential clinical impacts on treatments for early stage ALK positive small-cell lung cancer. The focus of the paper is well phrased. It is a useful commentary that provides a good overview of the latest findings in the perioperative alectinib treatment of ALK-positive lung cancer. I recommend that the paper is acceptable for publication after some minor and major revisions. The authors should also correct the points listed below.

Major comments

#1 The authors’ manuscript features the perioperative treatment with alectinib. To make the manuscript more balanced, it is desirable to briefly mention the ALCHEMIST trial (NCT02201992), an ongoing phase III trial examining the efficacy of preoperative crizotinib treatment in early stage ALK-positive lung cancer.

#2 I think that the findings of previous neoadjuvant treatments with ALK-TKIs for ALK-positive lung cancer should also be mentioned. Please review the results of previous phase II trials examining the efficacy neoadjuvant therapy with ALK-TKIs other than alectinib in the paragraph “3. Future perspectives” [e.g. SAKULA (UMIN00017906), ARM (NCT03088930)].  

Minor comments

#1 The authors should add data on the incidence of pneumonitis induced by alectinib adjuvant therapy after the sentence on line 81 of p2, referring to the Supplementary Table 5 in the ALINA study article published in the NEJM.

#2 The authors state “In the near future, the results of the ALNEO and NAUTIKA1 trials may further “clarify” the role of neoadjuvant/perioperative alectinib in the treatment paradigm of ALK-positive NSCLC patients.” in the Conclusion. However, the both trials are just phase II, or exploratory studies. Therefore, the authors cannot really say that the phase II trials "clarify" the role of neoadjuvant/perioperative alectinib. Please correct the term “clarify” in their text and replace it by “explore” or “examine”.

#3 The authors should add the start and end dates of the NAUTIKA1 and ALNEO trials in the text. Moreover, please describe the design of these trials in more detail in Figure 1 (e.g., phases of studies, single-center or multicenter, primary endpoint, and target enrollment).

#4 Please add the primary endpoint of each study and the expected start and end dates of the both trials in Figure 1.

#5 Please add the number and percentage of patients in the ALINA study who required discontinuation or dose reduction of alectinib postoperative treatment in the paragraphs describing safety and toxicity.

#6 What treatment did 15 patients receive when they relapsed after alectinib postoperative therapy? Please describe the subsequent treatments after the sentence on line 170 of p5, referring to the Supplementary Table 4 in the ALINA study article published in the NEJM.

#7 In the Abstract Results paragraph, it should be clearly stated that the ALINA trial showed positive results in “anaplastic lymphoma kinase (ALK)-positive” resected lung cancer.

#8 Please change the abbreviation of the gene mutation to italics throughout the manuscript (e.g., ALK instead of ALK).

#9 The authors should mention in the text that FDA approved alectinib as adjuvant treatment for ALK-positive non-small cell lung cancer on April, 18, 2024, and EMA on April, 25, 2024.

#10 Please add the reference number to Table 2.

#11 The term “Confidence Interval” on line 62 of p2 should be in all lowercase.

#12 The authors should provide a reference for the frequency of STAS in ALK-positive lung cancer (e.g. PMID 30089582).

#13 With respect to Figure 1 and the text, aren't the eligibility criteria for NAUTIKA1 trial stage IB-IIIA, selected IIIB? Please confirm. And please state what edition of the UICC these stages are based on.

I hope these comments will be helpful.

Author Response

Reviewer 1 comments:

This manuscript addresses potential clinical impacts on treatments for early stage ALK positive small-cell lung cancer. The focus of the paper is well phrased. It is a useful commentary that provides a good overview of the latest findings in the perioperative alectinib treatment of ALK-positive lung cancer. I recommend that the paper is acceptable for publication after some minor and major revisions. The authors should also correct the points listed below.

Comment #1: The authors’ manuscript features the perioperative treatment with alectinib. To make the manuscript more balanced, it is desirable to briefly mention the ALCHEMIST trial (NCT02201992), an ongoing phase III trial examining the efficacy of preoperative crizotinib treatment in early stage ALK-positive lung cancer.

Reply #1: Thank you. We appreciated your comments and we tried to improve our manuscript by mentioning the ALCHEMIST trial.

Comment #2: I think that the findings of previous neoadjuvant treatments with ALK-TKIs for ALK-positive lung cancer should also be mentioned. Please review the results of previous phase II trials examining the efficacy neoadjuvant therapy with ALK-TKIs other than alectinib in the paragraph “3. Future perspectives” [e.g. SAKULA (UMIN00017906), ARM (NCT03088930)].  

Reply #1: Thank you. We appreciated your comments and we tried to improve our manuscript with the results of SAKULA trial and a comment on ARM trial.

Minor comments

Comment #1: The authors should add data on the incidence of pneumonitis induced by alectinib adjuvant therapy after the sentence on line 81 of p2, referring to the Supplementary Table 5 in the ALINA study article published in the NEJM.

Reply #1: Thank you. We appreciated your comments and we tried to improve our manuscript with data on the incidence of pneumonitis.

Comment #2: The authors state “In the near future, the results of the ALNEO and NAUTIKA1 trials may further “clarify” the role of neoadjuvant/perioperative alectinib in the treatment paradigm of ALK-positive NSCLC patients.” in the Conclusion. However, the both trials are just phase II, or exploratory studies. Therefore, the authors cannot really say that the phase II trials "clarify" the role of neoadjuvant/perioperative alectinib. Please correct the term “clarify” in their text and replace it by “explore” or “examine”.

Reply #2: Thank you. We modified our manuscript.

Comment #3: The authors should add the start and end dates of the NAUTIKA1 and ALNEO trials in the text. Moreover, please describe the design of these trials in more detail in Figure 1 (e.g., phases of studies, single-center or multicenter, primary endpoint, and target enrollment).

Reply #3: Thank you. We appreciated your comments and we tried to improve our manuscript with the characteristics of the trials.

Comment #4: Please add the primary endpoint of each study and the expected start and end dates of the both trials in Figure 1.

Reply #4: Thank you. We appreciated your comments and we tried to improve our manuscript.

Comment #5: Please add the number and percentage of patients in the ALINA study who required discontinuation or dose reduction of alectinib postoperative treatment in the paragraphs describing safety and toxicity.

Reply #5: Thank you. We appreciated your comments and we tried to improve our manuscript with these data.

Comment #6: What treatment did 15 patients receive when they relapsed after alectinib postoperative therapy? Please describe the subsequent treatments after the sentence on line 170 of p5, referring to the Supplementary Table 4 in the ALINA study article published in the NEJM.

Reply #6: Thank you. We appreciated your comments and we tried to improve our manuscript with subsequent treatments.

Comment #7: In the Abstract Results paragraph, it should be clearly stated that the ALINA trial showed positive results in “anaplastic lymphoma kinase (ALK)-positive” resected lung cancer.

Reply #7: Thank you. We modified our manuscript.

Comment #8: Please change the abbreviation of the gene mutation to italics throughout the manuscript (e.g., ALK instead of ALK).

Reply #8: Thank you. We modified our manuscript.

Comment #9: The authors should mention in the text that FDA approved alectinib as adjuvant treatment for ALK-positive non-small cell lung cancer on April, 18, 2024, and EMA on April, 25, 2024.

Reply #9: Thank you. We appreciated your comments and we tried to improve our manuscript with FDA and EMA approval.

Comment #10: Please add the reference number to Table 2.

Reply #10: Thank you. We modified our manuscript.

Comment #11: The term “Confidence Interval” on line 62 of p2 should be in all lowercase.

Reply #11: Thank you. We modified our manuscript.

Comment #12: The authors should provide a reference for the frequency of STAS in ALK-positive lung cancer (e.g. PMID 30089582).

Reply #12: Thank you. We appreciated your comments and we tried to improve our manuscript with this reference.

Comment #13: With respect to Figure 1 and the text, aren't the eligibility criteria for NAUTIKA1 trial stage IB-IIIA, selected IIIB? Please confirm. And please state what edition of the UICC these stages are based on.

Reply #13: We have reported the disease stages described in the methods of the abstract included in the references.

Lee, J.; Sepesi, B.; Toloza, E.; Lin, J.; Pass, H.; Johnson, B.; Heymach, J.; Johnson, M.; Ding, B.; Schulze, K.; et al. EP02.04-005 Phase II NAUTIKA1 Study of Targeted Therapies in Stage II-III NSCLC: Preliminary Data of Neoadjuvant Alectinib for ALK+ NSCLC. J. Thorac. Oncol. 2022, 17, S233–S234, https://doi.org/10.1016/j.jtho.2022.07.390.

Reviewer 2 Report

Comments and Suggestions for Authors

The authors comment in their work on clinical studies of a new standard in the treatment of Non-small cell lung cancer; At the same time, the use of alectinib as a new drug for the treatment of this type of cancer with Alterations of the anaplastic lymphoma kinase is being considered. The authors reviewed studies on this topic in recent years and concluded that adjuvant alectinib represents the new standard of care and could soon be used in perioperative treatment.

For a better understanding of the topic of this commentary, the authors should add a description of this adjuvant to the work, and give a comparative description of this drug in comparison with analogues.

In addition, it is necessary to illustrate with a diagram/scheme the possibilities of this therapy and include several figures from original articles for a better presentation of the commentary material

Author Response

Reviewer 2 comments:

The authors comment in their work on clinical studies of a new standard in the treatment of Non-small cell lung cancer; At the same time, the use of alectinib as a new drug for the treatment of this type of cancer with Alterations of the anaplastic lymphoma kinase is being considered. The authors reviewed studies on this topic in recent years and concluded that adjuvant alectinib represents the new standard of care and could soon be used in perioperative treatment.

Comment #1: For a better understanding of the topic of this commentary, the authors should add a description of this adjuvant to the work, and give a comparative description of this drug in comparison with analogues.

Reply #1: Thank you. We appreciated your comments and we tried to improve our manuscript with trials that evaluated crizotinib and ceritinib.

Comment #2: In addition, it is necessary to illustrate with a diagram/scheme the possibilities of this therapy and include several figures from original articles for a better presentation of the commentary material

Reply #2: We created figures and tables included in the paper to best summarize the data reported and we do not consider it necessary to include figures or tables already published.

Reviewer 3 Report

Comments and Suggestions for Authors

It was my pleasure to review this short commentary titled “Alectinib in early-stage ALK-positive NSCLC: current evidence and future challenges.” by Dr Diego L. Cortinovis and his colleagues.

In this report the authors present a brief overview of two recent trials utilizing targeted tyrosine kinase inhibitors (TKI) for ALK-re-arranged NSCLC.

This is a nice little review/overview of the thus far progress on the management of these patients. It is well written and presented with clear and informative images and tables.

I would be happy to recommend its publication.

Kind regards to all.

Detail comments

Check spelling of ALK-rearrenged NSCLC word!

Author Response

It was my pleasure to review this short commentary titled “Alectinib in early-stage ALK-positive NSCLC: current evidence and future challenges.” by Dr Diego L. Cortinovis and his colleagues. In this report the authors present a brief overview of two recent trials utilizing targeted tyrosine kinase inhibitors (TKI) for ALK-re-arranged NSCLC. This is a nice little review/overview of the thus far progress on the management of these patients. It is well written and presented with clear and informative images and tables. I would be happy to recommend its publication.

Detail comments

Comment #1: Check spelling of ALK-rearrenged NSCLC word!

Reply #1: Thank you. We modified our manuscript.

Reviewer 4 Report

Comments and Suggestions for Authors

It is interesting that in lines 55-58 you note the overrepresentation of females and smokers in the alectinib-treated group.  I am concerned that this allocation of research subjects might have biased the outcome to increase the benefit attributed to alectinib.  This possibility should at least be acknowledged in the Discussion section.

Author Response

Reviewer 4 comments:

Comment #1: It is interesting that in lines 55-58 you note the overrepresentation of females and smokers in the alectinib-treated group.  I am concerned that this allocation of research subjects might have biased the outcome to increase the benefit attributed to alectinib.  This possibility should at least be acknowledged in the Discussion section.

Reply #1: Thank you. We appreciated your comments and we tried to improve our manuscript in the discussion.